# Future Perspective for ALK-Positive Anaplastic Large Cell Lymphoma with Initial Central Nervous System (CNS) Involvement: Could Next-Generation ALK Inhibitors Replace Brain Radiotherapy for the Prevention of Further CNS Relapse?

Makito Tanaka [1,*], Hiroki Miura [1], Soichiro Ishimaru [1], Gen Furukawa [1], Yoshiki Kawamura [1], Kei Kozawa [1], Seiji Yamada [2], Fumitaka Ito [3], Kazuko Kudo [1] and Tetsushi Yoshikawa [1]

1   Department of Pediatrics, Fujita Health University School of Medicine, 1-98 Dengakugakubo, Kutsukake-cho, Toyoake 470-1192, Aichi, Japan
2   Department of Diagnostic Pathology, Fujita Health University School of Medicine, Toyoake 470-1192, Aichi, Japan
3   Department of Radiation Oncology, Fujita Health University School of Medicine, Toyoake 470-1192, Aichi, Japan
*   Correspondence: makito-t@fujita-hu.ac.jp; Tel.: +81-562-93-9251; Fax: +81-562-95-2216

**Abstract:** Central nervous system (CNS) involvement in anaplastic large cell lymphoma (ALCL) at diagnosis is rare and leads to poor prognosis with the use of the standard ALCL99 protocol alone. CNS-directed intensive chemotherapy, such as an increased dose of intravenous MTX, increased dose of dexamethasone, intensified intrathecal therapy, and high-dose cytarabine, followed by cranial irradiation, has been shown to improve survival in this population. In this paper, the authors describe a 14-year-old male with an intracranial ALCL mass at onset who received CNS-directed chemotherapy followed by 23.4 Gy of whole-brain irradiation. After the first systemic relapse, the CNS-penetrating ALK inhibitor, alectinib, was applied; it has successfully maintained remission for 18 months without any adverse events. CNS-penetrating ALK inhibitor therapy might prevent CNS relapse in pediatric ALK-positive ALCL. Next-generation ALK inhibitors could be introduced as a promising treatment option, even for primary ALCL with CNS involvement, which could lead to the omission of cranial irradiation and avoid radiation-induced sequalae. Further evidence of CNS-penetrating ALK inhibitor combined therapy for primary ALK-positive ALCL is warranted to reduce radiation-induced sequalae in future treatments.

**Keywords:** anaplastic large cell lymphoma; central nervous system; ALK inhibitor; alectinib; irradiation; children

## 1. Introduction

Anaplastic large cell lymphoma (ALCL) is a mature T-cell non-Hodgkin lymphoma that accounts for 15% of childhood non-Hodgkin lymphomas [1,2]. The international ALCL99 study is the largest phase III clinical trial for pediatric ALCL, and this treatment is considered to be the current standard therapy [3]. Pediatric patients with newly diagnosed anaplastic lymphoma kinase (ALK)-positive ALCL treated with the ALCL99 treatment regimen have 65–75% event-free survival (EFS).

Central nervous system (CNS) involvement in ALCL diagnosis is rare (2–3% of primary ALCL) [4]. Initial CNS involvement in ALCL was also registered in the ALCL99 cohort, but its treatment was not defined in its protocol and was heterogeneous [5]. The largest multicenter European and Japanese retrospective study for childhood ALCL with CNS involvement examined the treatment and outcomes of 13 patients, wherein CNS-directed intensive chemotherapy followed by cranial irradiation was mainly applied [6]. Either LMB

group C or BFM-NHL-90 K3 regimen, which is based on the short-pulsed B-cell lymphoma strategy [7,8], were used as CNS-directed chemotherapy [6,9].

However, it is well known that cranial irradiation is associated with a potential risk of various radiation-induced complications, such as intellectual or neurocognitive dysfunction, endocrinological deficit, secondary malignant neoplasm, cataracts, and glaucoma [10,11]. These radiation-related adverse toxicities affect the long-term sequelae for children during their development, and irradiation at a younger age is significantly correlated with the severity of complications. Therefore, a less toxic but highly effective treatment strategy, if possible without cranial irradiation, is desirable for childhood ALCL with CNS involvement [12].

The World Health Organization (WHO) classified ALCL into four categories: ALK-positive ALCL, ALK-negative ALCL, primary cutaneous ALCL, and breast-implant-associated ALCL [13]. Over 90% of childhood ALCL harbors the translocation of the *ALK* gene at 2p23, which leads to constitutive ALK-kinase activation. Recent reports have shown the efficacy of ALK inhibitors for relapsed or refractory ALK-positive ALCL with less toxicity. The toxicity profile of next-generation ALK inhibitors has been extensively studied to address ALK-positive non-small cell lung cancer (NSCLC) [14,15]. A systematic review revealed that the common adverse events (AEs) observed with ALK inhibitors were gastrointestinal toxicities, such as nausea, vomiting and diarrhea, elevation of liver enzymes and fatigue. Most of the AEs were low grade and treatment-related deaths were associated with ALK inhibitors in less than 1% of patients [16]. Among ALK inhibitors, alectinib has shown superior safety through meta-analysis [17]. Similar toxicity profiles of ALK inhibitors were reported on smaller studies on ALK-positive ALCL in children [18,19]. Regarding the efficacy of ALK inhibitors, while the first-generation ALK inhibitor, crizotinib, shows less CNS penetrance, next-generation ALK inhibitors, such as alectinib, brigatinib, ceritinib, or lorlatinib, have been developed and can reach the CNS lesion through the blood–brain barrier. Several reports suggested the clinical significance of the next-generation ALK inhibitors, mainly in relapsed or refractory settings [20–22], but there is not sufficient evidence in primary settings.

Here, we describe a 14-year-old male with an intracranial ALCL mass at diagnosis, who was treated with an NHL-BFM-90 K3 regimen as the CNS-directed intensive chemotherapy, followed by 23.4 Gy of whole-brain irradiation. This regimen contains an increased dose of intravenous MTX, increased dose of dexamethasone, intensified intrathecal therapy, and high-dose cytarabine, compared to the standard ALCL99 protocol. As we lacked the evidence of CNS-penetrating next-generation ALK inhibitors against initial CNS involvement of ALCL during the initial diagnosis, we applied the conventional cranial irradiation following the CNS-directed chemotherapy for this patient.

Patients with CNS disease often receive cranial radiation. Cranial radiation is a risk factor for both short- and long-term effects, especially in children. Next-generation ALK inhibitors cross the blood–brain barrier, and thus in the future we might be able to use these drugs to avoid radiation in patients with CNS disease and potentially prevent CNS relapses for those without CNS disease. The case report of the patient could provide an illustration of the dilemma of using cranial radiation in an age when ALK inhibitors exist that cross the blood–brain barrier.

## 2. Case Description

A 14-year-old male had complained of occasional blinking and transient visual deficit on the right side for several weeks. He developed afebrile convulsions with no apparent epileptic discharge on electroencephalography. Cranial magnetic resonance imaging (MRI) revealed an isolated mass with a hyperintense signal in the left parietal lobe on T2- and diffusion-weighted images (Figure 1A–C). A systemic contrast CT scan did not show any other lesions. Repeated cranial MRI over two weeks showed the progression of a hyperintense signal on the gadolinium-enhanced image in the left parietal lesion and surrounding hyperintense signal on the T2-weighted image, suggesting a brain tu-

mor and surrounding edema. A biopsy of the brain lesion was performed. The patient was pathologically diagnosed with ALK-positive ALCL. Large pleomorphic cells with lobulated nuclei were diffusely proliferated with frequent mitosis, which occasionally contained horseshoe-shaped nuclei. Immunostaining on these atypical large cells was positive for CD30, granzyme, EMA, ALK1, CD3, and CD4 and negative for CD20, CD8, and CD56 (Figure 2A–D). The MIB-1 index was around 80%, and EBER in situ hybridization was negative.

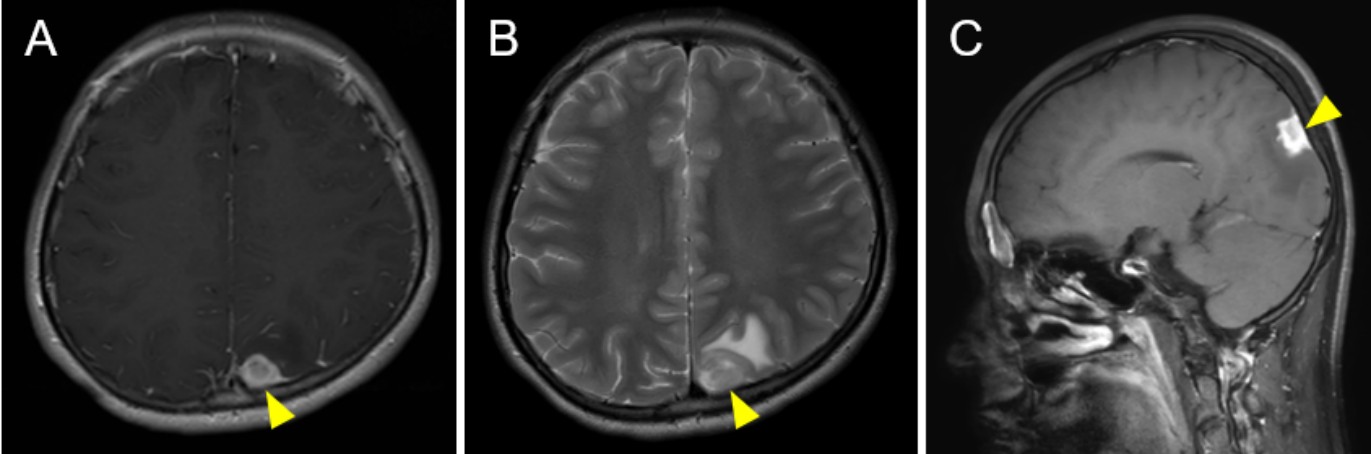

**Figure 1.** Cranial magnetic resonance imaging (MRI) before brain biopsy demonstrated (**A**) a hypointense tumor (arrowhead) in the left parietal lobe on T1-weighted imaging that was (**B**) surrounded by hyperintense edema (arrowhead) on T2-weighted imaging. (**C**) A contrast-enhanced T1-weighted sagittal image after brain biopsy showed diffuse enhancement of the tumor surrounded by unenhanced edema.

After referral to our hospital, a positron emission tomography-CT (PET-CT) scan showed several small systemic lesions in his liver, lung, and cervical, axillary, and para-aortic lymph nodes, which suggested subsequent disseminated disease. Initially, he received the first course from the ALCL99 protocol (5-day prophase and course AM), including intravenous methotrexate (MTX) 3 g/m$^2$ for 3 h. Severe progressive headache and vomiting disappeared after the first course of chemotherapy. He subsequently received five courses of the CNS-directed NHL-BFM-90 K3 regimen. Vital points for CNS-directed therapy, whose courses and dosages from the NHL-BFM-90 regimen K3 were applied to this patient, are given below. He received a 5-day cytoreductive prophase: dexamethasone 5 mg/m$^2$ on days 1 and 2; dexamethasone 10 mg/m$^2$ on days 3, 4, and 5; and cyclophosphamide 200 mg/m$^2$ on days 1 and 2. After this prephase therapy, the patient stratified to the course AA, followed by courses BB, CC, AA, BB, and CC. In the therapy of courses AA and BB, MTX 5 g/m$^2$ was administered for 24 h with subsequent leucovorin rescue. Dexamethasone 10 mg/m$^2$ for 5 days is included in courses AA and BB while dexamethasone 20 mg/m$^2$ for 5 days is included in course CC. Twice-daily intravenous administration of high-dose cytarabine 2 g/m$^2$ on days 1 and 2 is included in course CC [3]. Complete remission was achieved after the initial two courses of chemotherapy. As CNS consolidation therapy, he received 23.4 Gy of whole-brain irradiation after these six courses of chemotherapy.

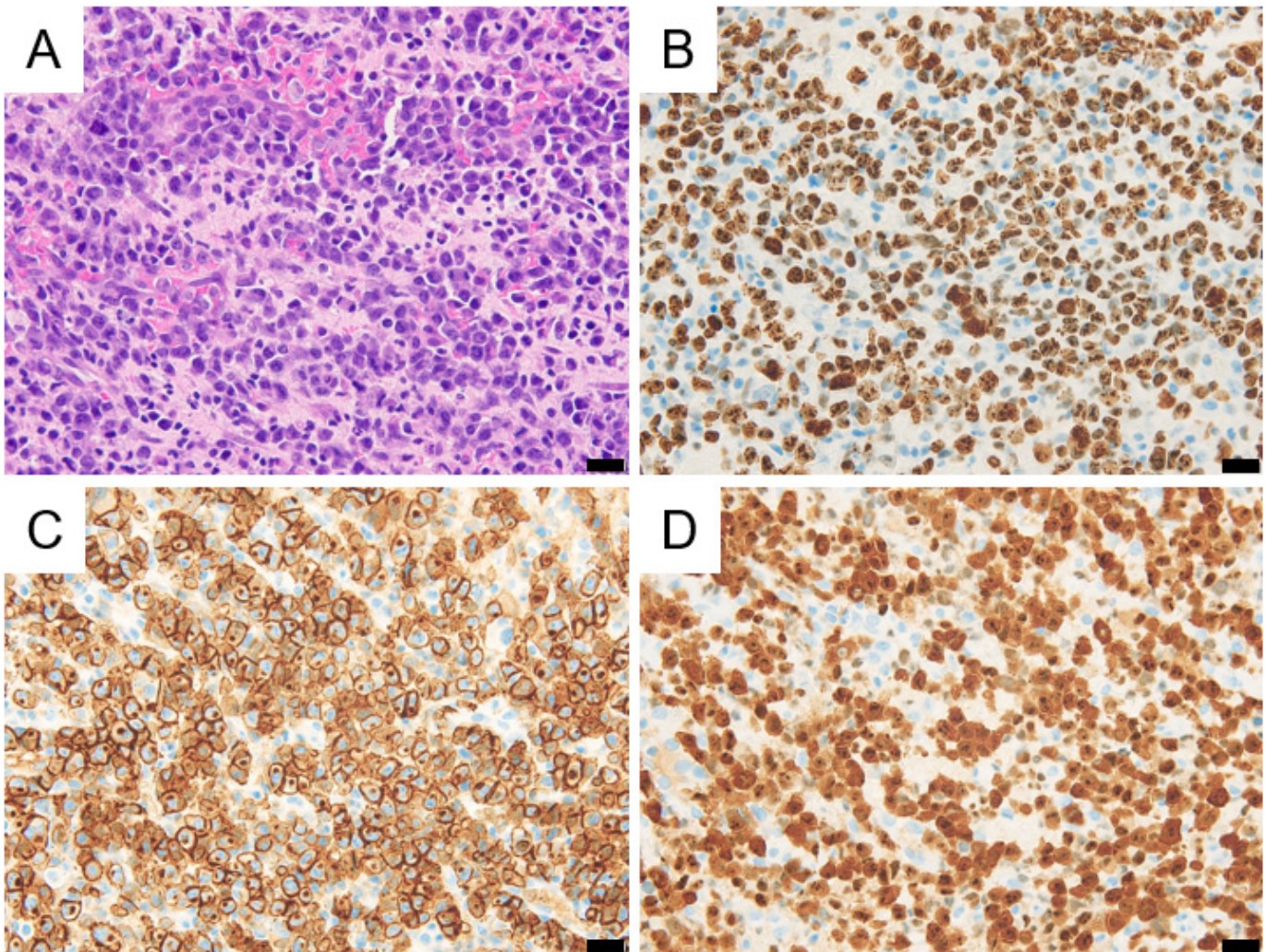

**Figure 2.** (**A**) Hematoxylin–eosin stain of a brain biopsy sample revealed proliferation of large atypical lymphoid cells with densely lobulated nuclei and frequent mitosis. (**B**) MIB-1 immunohistochemical staining showed 80% positivity. Neoplastic cells expressed CD30 in the cytoplasmic membrane (**C**) and ALK in the nucleus and the cytoplasm (**D**). Black bars denote 20 μm.

At 8 months after treatment completion, however, the patient returned to our hospital with fever, skin rash, and lymphadenopathy. He showed a significant elevated level of the soluble IL-2 receptor at $8.2 \times 10^4$ IU/L. Skin and cervical lymph node biopsies showed a diffuse proliferation of large CD30- and ALK-positive "hallmark" cells, and the tumor was positive for NPM-ALK fusion on reverse transcriptase–polymerase chain reaction [23]. Thus, he was diagnosed with recurrent ALK-positive ALCL. Contrast CT and PET-CT scans showed systemic dissemination in the skin, lung, and lymph nodes. Since the cytology of cerebrospinal fluid was negative and the head MRI showed no CNS lesions (data not shown), we considered him to have systemic relapse without CNS involvement. Respiratory distress due to lung metastasis and pleural effusion, even after preceding dexamethasone treatment, required immediate chemotherapy. Complete remission was achieved after the first course of ALCL99 chemotherapy, which consisted of the 5-day prophase following course AM, including high-dose methotrexate. Subsequently, he received second-generation ALK inhibitor maintenance, alectinib at 300 mg twice daily, without any adverse events, and he has remained in remission for 18 months from the start of the alectinib treatment. Now, he is a senior high school student who undergoes daily oral alectinib administration and regular medical checkups in an outpatient clinic.

Written informed consent for this publication was obtained from the participating patient and his guardians, following the rules of the Declaration of Helsinki.

## 3. Discussion

CNS involvement is associated with poor prognosis in ALCL. This suggests that CNS-directed intensive chemotherapy penetrating the blood–brain barrier (BBB) is required to improve the prognosis of ALCL with CNS involvement. Del Baldo et al. reported that only one out of five CNS-relapsed patients after primary CNS involvement survived [21]. Since rescue after CNS relapse is difficult, the importance of intensified therapy for primary CNS involvement has been suggested [21,22]. The largest retrospective review of primary childhood ALCL with CNS involvement showed that NHL-BFM-90/95-based chemotherapy combined with whole-brain irradiation is mainly used as the treatment [6,24]. Thorer et al. reported five CNS-involved ALCL patients who received 24 Gy whole-brain irradiation after chemotherapy. Three patients are alive in remission, while one died due to infection and another died after relapse [25].

At the initial diagnosis of our patient, we applied NHL-BFM-90-based CNS-directed chemotherapy following 23.4 Gy whole-brain irradiation. After systemic relapse, following the re-induction chemotherapy, a CNS-penetrating ALK inhibitor, alectinib, was given to the patient, and he has been in complete remission for 18 months. Alectinib treatment after systemic relapse for our primary CNS-involved ALCL patient does not directly show the effectiveness of CNS prophylaxis. However, it could possibly support the prevention of CNS relapse for this patient, based on the clinical trial of NSCLC and case reports on ALCL.

Recent advances in targeted therapy provide several treatment options for relapsed or refractory pediatric ALCL, including CD30-targeted brentuximab vedotin (BV) and ALK-targeted therapy [26]. Ruf et al. reported five CNS-relapse cases during BV and vinblastine (VBL) therapy for primary ALCL, which suggested that BV or VBL has less efficacy against CNS lesions [27].

ALK, a tyrosine kinase, is an important oncogenic driver of ALCL [28]. It is expressed in over 90% of pediatric ALCL cases [29]; it is also considered to be a good therapeutic target. Over the past few decades, ALK inhibitors have brought about certain clinical benefits for various cancers. However, kinase selectivity, localization, and drug resistance are crucial problems for sustained clinical efficacy to apply the disease-specific ALK inhibitor [30].

Crizotinib is the first FDA-approved ALK inhibitor, which, unfortunately, faces rapid drug resistance [31]. To overcome this problem, ceritinib, alectinib, and brigatinib (second-generation inhibitors) or lorlatinib (third-generation inhibitor) have been developed [18].

Several next-generation ALK inhibitors have been demonstrated to cross the BBB and have good CNS penetration [31]. In comparison to first-generation crizotinib, alectinib, has excellent CNS penetration [32,33] and is an appropriate candidate for CNS-directed therapy. Through the phase III randomized trial for ALK-rearranged untreated NSCLC, the J-ALEX study, alectinib has proven superior efficiency over crizotinib for preventing CNS metastasis [34]. Del Baldo et al. reported two CNS-relapsed ALCL patients who were successfully treated with ceritinib or alectinib. Recently, Rigaud et al. reported ten relapsed/refractory ALCL patients with CNS involvement; nine patients were alive after next-generation ALK inhibitor treatment [22].

The introduction of a next-generation ALK inhibitor to the primary CNS-involved ALCL could help avoid whole-brain irradiation and its sequalae. This has a significant impact on younger ALCL patients. Since the frequency of ALK-positive ALCL patients with CNS involvement is rare, an international collaboration is expected to enable the elucidation of the safety and efficacy of CNS-penetrating next-generation ALK inhibitor treatment. ALK inhibitor monotherapy is reported to lead to disease recurrence after the cessation of ALK inhibitors, thus requiring prolonged ALK inhibitor administration [35]. Aroiso et al. demonstrated that simultaneous combinations, not the sequential combinations, of ALK inhibitors and chemotherapy confirmed their efficacy in inhibiting tumor cell growth. They

proposed that upfront combined treatments prevent the selection of resistant clones in ALK-positive ALCL cells from in vitro and murine model experiments [36].

The integration of ALK inhibitors with cytotoxic chemotherapy is of utmost importance for rare CNS-involved ALCL [18,37]. Crizotinib in combination with standard chemotherapy in front-line ALK-positive ALCL treatment was assessed in the Children's Oncology Group trial ANHL12P1 (ClinicalTrials.gov identifier: NCT01979536) [38]. This combination therapy showed a relatively low relapse rate; the 2-year event-free survival and overall survival were 76.8% and 95.2%, respectively. Recently, He et al. reported four primary systemic ALCL patients who were treated with alectinib-included ALCL-99 treatment; the first two cycles led to complete remission in all four patients [39]. Yang et al. reported that the simultaneous administration of alectinib and chemotherapy showed significant and durable CNS effects in a relapsed pediatric ALK+ ALCL patient [40]. Accumulated evidence for the safety and efficacy of next-generation ALK inhibitors simultaneously administered with chemotherapy is warranted for primary CNS-involved ALCL treatment.

### 4. Conclusions

Newly diagnosed ALCL with CNS involvement has often been treated with CNS-directed chemotherapy and whole-brain irradiation. Next-generation ALK inhibitors have shown excellent transfer efficiency across the BBB and has also been proven to be quite safe, thus it could be a promising treatment option for newly diagnosed ALCL with CNS involvement, eventually leading to the avoidance of radiation-induced sequalae. This benefit would be reserved for younger children. A large-scale international clinical trial using this CNS-penetrating ALK inhibitor-combined chemotherapy for primary CNS-involved ALK-positive ALCL is warranted to prove the efficacy of this treatment option.

**Author Contributions:** M.T. managed the patient and wrote the manuscript. T.Y. supervised and edited the manuscript. H.M., S.I., G.F., Y.K., K.K. (Kei Kozawaand), S.Y., F.I. and K.K. (Kazuko Kudo) managed the patient and revised the manuscript. All authors have read and agreed to the published version of the manuscript.

**Funding:** This research received no external funding.

**Institutional Review Board Statement:** Not applicable.

**Informed Consent Statement:** Written informed consent was obtained from the patient and his guardian to publish this paper.

**Data Availability Statement:** Not applicable.

**Acknowledgments:** The authors thank Yuka Iijima for their skillful assistance in measuring the NPM-ALK mRNA.

**Conflicts of Interest:** The authors declare no conflict of interest.

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
