# Peer review of "Future Perspective for ALK-Positive Anaplastic Large Cell Lymphoma with Initial Central Nervous System (CNS) Involvement: Could Next-Generation ALK Inhibitors Replace Brain Radiotherapy for the Prevention of Further CNS Relapse?"

_pediatrrep, doi:10.3390/pediatric15020029_

Round 1

Reviewer 1 Report

The article "Future Perspective for ALK-positive Anaplastic Large Cell Lymphoma with Initial Central Nervous System (CNS) Involvement: Could Next Generation ALK Inhibitors Replace Brain Radiotherapy for Prevention of Further CNS Relapse?" by Tanaka et al. describes a case of ALK+ ALCL that was CNS positive at diagnosis. The authors describe the initial treatment which included cranial radiation. The patient then relapsed 8 months later and has been treated with alectinib and continues in CR. The patient did not have CNS involvement at relapse. Overall, the authors bring up two important clinical questions in ALK+ ALCL: Can the use of ALK inhibitors such as alectinib treat CNS disease in ALCL without the use of radiation? Can the use of ALK inhibitors such as alectinib prevent CNS recurrence? The discussion of these two questions is of interest to oncologists. That said, the article seems to try and fit a case report to these questions when the case doesn’t have much of a relation to either question. The patient was CNS positive at diagnosis and treated with radiation, not alectinib, and at relapse the patient was CNS negative. In addition, patients who had CNS disease at diagnosis and then relapsed often do not have CNS disease at relapse making it impossible to tell even anecdotally whether alectinib had any effect on preventing CNS relapse. It is the opinion of this reviewer that the article would be strengthened by simply stating that: patients with CNS disease often receive cranial radiation, cranial radiation is a risk factor for short and long term effects especially in children, ALK inhibitors cross the blood brain barrier, and the future might allow us to use these drugs to avoid radiation in patients with CNS disease and potentially prevent CNS relapses for those without CNS disease. The case report of the patient could be more of an illustration of the dilemma of using cranial radiation in an age when ALK inhibitors exist that cross the blood brain barrier.

A few specific comments:

·       Lines 31-33 describing short-pulsed B-cell therapy do not add anything to the report and could be deleted.

·       Lines 38-48 are simply a paragraph describing reference 5 (referenced 3 separate times). The paragraph should be simplified as a review of the reference.

·       Line 46 discusses a survival benefit but does not describe the comparator group.

·       Line 47 discusses recommended cranial radiation dose but does not reference who recommended this dose

·       Line 52-54 can be deleted as mentioning that risks should be discussed during treatment options has nothing to do with the case. If the authors would like, a discussion of potential risks of radiation versus the unknown long term risks of ALK inhibitors would be appropriate for the discussion.

·       Line 73 the authors discuss that “whole brain irradiation at his age would have acceptable side effects” – aren’t the side effects of radiation, particularly the long term effects, the reason the authors discuss the use of alectinib over radiation? If the effects of radiation were completely acceptable then the article doesn’t have to be written in the first place.

·       Lines 67-82 describe the patient but then the case report of the patient follows. There is a lot of redundancy between the two sections.

·       Overall the discussion brings up many good points but suffers from English syntax which is likely secondary to English being a second language. Authors should consider a service to help with language clarity.

Reviewer 2 Report

Dear Editor and authors,

I have carefully read the manuscript concerning possible treatment or prevention of CNS involvement with next generation ALK Inhibitors in ALK-positive anaplastic large cell pediatric lymphoma.

In my opinion, this case report is correctly and interestingly discussed.

However, text editing for repetitions would improve the consistency and clarity of the manuscript and should be considered.

The therapeutic implications of second-generation ALK-Inhibitors, such as alectinib and ceritinib, are currently evaluated in several pediatric tumours, such as neuroblastoma, anaplastic large-cell lymphoma and CNS tumours. Some first single-centre data with alectinib has already been published regarding this type of pediatric lymphoma. The manuscript is interesting because it refers to the potential therapeutic or recurrence-preventive role of alectinib in the case of a rare CNS involvement in ALK-positive anaplastic large-cell pediatric lymphoma. The images in the manuscript are appropriate to confirm the CNS involvement. The references are also appropriate.

The main shortcoming of the manuscript is that the authors’ hypothesis, although of clinical impact, is primarily supported by the critical evaluation of available published data rather than the case, which is not typical. The ALK-targeted inhibition has been used for systemic recurrence and not for the primary disease, including CNS involvement.

Lastly, if the manuscript is accepted for revision following my evaluation, the text will require editing before publication in order to eliminate the repetitions concerning the case (e.g. both in the introduction and the case report section) and the ALK-Inhibition-related data.

Reviewer 3 Report

In general, this is an interesting case-report raising the issue of the treatment of CNS involvement in ALCL.

I have some minor comments:

- in lines 61-62 - please explain the phrase "less acceptable toxicity" (here or in the discussion part); it is especially important as you generally claim in the paper, that ALK inhibitors would have less toxicities than CNS radiotherapy.

- the clinical trial mentioned in lines 194-196 (NCT04925609) is meant for refractory/relapsed ALCL only, so it will not directly answer the question of the efficacy of ALK inhibitor in the ALCL with primary CNS involvement

- some paragraphs in the discussion part are too detailed and not directly related to the topic (e.g. lines 149-156 could be shortened substantially)

- conclusions are very hard to understand due to grammatical errors. Please re-write this part with the help of a native speaker 

- in general, I would recommend proofreading the article by a native speaker. The English level in most of the sentences is fair, however, there are some parts that are difficult to understand.

Round 2

Reviewer 1 Report

The authors have addressed a number of this reviewer's points which are greatly appreciated. There are still a few items that could be improved which would increase the readability of the manuscript.

Line 38-39: This isn't a sentence. Need to reword.

Line 45: "recommended" should be changed to "used" since the regimen was not recommended but rather the treatment is what was used by the physicians.

Line 57: I do not know what "less acceptable toxicity" means? Please clarify.

Line 60-63: The sentences dont flow and dont connect. Does the last sentence refer to the case without any intro of the case? Lines need to be clarified.

Line 68-69: "by comparing to" should be "compared to"

Line 95 and 116: The sIL2 levels have nothing to add to the discussion and can be omitted.

Line 133: There is no evidence that CNS relapse is difficult to treat (or if there is specific reference then the authors should reference it). Suggest changing.

Line 136: As mentioned previously, the term "survival benefit" suggests benefit over something else. As the authors wrote in their response, this is difficult to show so would suggest not using that term and just stating what chemotherapy was used.
